# Prevalence of mental health conditions, sensory impairments and physical disability in people with co-occurring intellectual disabilities and autism compared with other people: a cross-sectional total population study in Scotland

Kirsty Dunn ,[1] Ewelina Rydzewska,[1] Michael Fleming,[2] Sally-Ann Cooper [1]

[1]Institute of Health and Wellbeing, University of Glasgow, Glasgow, Scotland, UK
[2]Department of Public Health, University of Glasgow, Glasgow, Scotland, UK

**Correspondence to**
Kirsty Dunn;
kirsty.wright@glasgow.ac.uk

## ABSTRACT

**Objectives** To investigate prevalence of mental health conditions, sensory impairments and physical disability in children, adults and older adults with co-occurring intellectual disabilities and autism, given its frequent co-occurrence, compared with the general population.
**Design** Whole country cohort study.
**Setting** General community.
**Participants** 5709 people with co-occurring intellectual disabilities and autism, compared with 5 289 694 other people.
**Outcome measures** Rates and ORs with 95% CIs for mental health conditions, visual impairment, hearing impairment and physical disability in people with co-occurring intellectual disabilities and autism compared with other people, adjusted for age, sex and interaction between age and co-occurring intellectual disabilities and autism.
**Results** All four long-term conditions were markedly more common in children, adults and older adults with co-occurring intellectual disabilities and autism compared with other people. For mental health, OR=130.8 (95% CI 117.1 to 146.1); visual impairment OR=65.9 (95% CI 58.7 to 73.9); hearing impairment OR=22.0 (95% CI 19.2 to 25.2); and physical disability OR=157.5 (95% CI 144.6 to 171.7). These ratios are also greater than previously reported for people with *either* intellectual disabilities *or* autism rather than co-occurring intellectual disabilities and autism.
**Conclusions** We have quantified the more than double disadvantage for people with co-occurring intellectual disabilities and autism, in terms of additional long-term health conditions. This may well impact on quality of life. It raises challenges for staff working with these people in view of additional complexity in assessments, diagnoses and interventions of additional health conditions, as sensory impairments and mental health conditions in particular, compound with the persons pre-existing communication and cognitive problems in this context. Planning is important, with staff being trained, equipped,

## Strengths and limitations of this study

► Large-scale, whole country study, with a high response rate (94%), so the results are representative of the whole population.
► Intellectual disabilities, autism and additional long-term conditions were enquired about systematically for everyone in the population.
► The wording of questions was tested in advance, via cognitive question testing during the design of Scotland's Census 2011.
► Limitations include proxy reporting.
► People known to have autism/Asperger's syndrome, intellectual disabilities and the four long-term conditions were reported, rather than each undergoing detailed individual research assessments that are not possible in such large population studies.

resourced and prepared to address the challenge of working for people with these conditions.

## BACKGROUND

People with intellectual disabilities[1–4] and people with autism[5–7] have more mental and physical health needs than other people. A whole population study using the Scotland's Census 2011 reported that 21.7% of people with intellectual disabilities also had autism and 18.0% of people with autism also had intellectual disabilities,[8] so this dually diagnosed group warrant investigation. One would suspect that this population with co-occurring intellectual disabilities and autisms likely to have a high level of additional health needs, but this has received little previous attention. A higher number of additional health needs increases the likelihood of misdiagnosis and

**Table 1** Characteristics of people with and without co-occurring intellectual disabilities and autism

| | People without co-occurring intellectual disabilities and autism n=5 289 694 (100%) Number (%) | People with co-occurring intellectual disabilities and autism n=5709 (100%) Number (%) |
|---|---|---|
| Gender* | | |
| Males | 2 563 675 (48.5) | 3769 (66.0) |
| Females | 2 726 019 (51.5) | 1940 (44.0) |
| Age groups* (years) | | |
| 0–15 | 913 969 (17.3) | 2362 (41.4) |
| 16–24 | 631 094 (11.9) | 1394 (24.4) |
| 25–34 | 666 725 (12.6) | 602 (10.5) |
| 35–44 | 734 304 (13.9) | 450 (7.9) |
| 45–54 | 786 355 (14.9) | 401 (7.0) |
| 55–64 | 667 157 (12.6) | 256 (4.5) |
| 65+ | 890 090 (16.8) | 244 (4.3) |
| Country of birth* | | |
| UK | 4 920 614 (93.0) | 5505 (96.4) |
| Other Europe | 172 160 (3.3) | 83 (1.5) |
| Africa | 46 708 (0.9) | 34 (0.6) |
| Middle East and Asia | 104 480 (2.0) | 50 (0.9) |
| The Americas and the Caribbean | 33 325 (0.6) | 28 (0.5) |
| Other | 12 407 (0.2) | 9 (0.2) |
| Ethnicity | | |
| White | 5 078 910 (96.0) | 5497 (96.3) |
| Asian | 140 542 (2.7) | 136 (2.4) |
| Mixed/multiple ethnicities | 19 775 (0.4) | 40 (0.7) |
| African | 29 615 (0.6) | 23 (0.4) |
| Caribbean or black | 6536 (0.1) | 4 (0.1) |
| Other ethnic groups | 14 316 (0.3) | 9 (0.2) |

*People with co-occurring intellectual disabilities and autism versus people without co-occurring intellectual disabilities and autism; p<0.01.

treatment interactions, so requires more complex treatment plans. Hence, it is important to investigate long-term additional health needs experienced by people with co-occurring intellectual disabilities and autism.

Some studies have investigated mental ill-health in people with co-occurring intellectual disabilities and autism. A small study of 149 adults with severe or profound intellectual disabilities and autism, living in state-run developmental centres in Louisiana, USA, compared coexisting conditions with 158 adults with intellectual disabilities without autism in the same centres. The former group had more symptomology for anxiety, mania, schizophrenia, stereotypes, self-injurious behaviour, eating disorders, sexual disorders and impulse control.[9] A study in Norway compared 62 adults with co-occurring autism and intellectual disabilities under the care of autism services, with 132

adults with intellectual disabilities only receiving intellectual disabilities support.[10] High levels of psychiatric disorders were reported in both groups: 53.2% in the co-occurring intellectual disabilities and autism group and 17.4% in the intellectual disabilities only group. An English study of referrals to a specialist intellectual disabilities psychiatric service described 42% of the 137 referred adults who had autism as well as intellectual disabilities to have coexisting psychopathology, most commonly schizophrenia.[11] A study of young people aged 14–20 years age, gender matched 36 people with co-occurring intellectual disabilities and autism with 36 people with intellectual disabilities without autism.[12] They reported the former group to have more episodes of mental ill-health, most commonly depression. A study of people aged 8–29 years with intellectual disabilities and challenging behaviour living in four residential units in England included 69 who also had autism and 13 who did not.[13] They reported a higher prevalence of organic disorders, anxiety and stereotypes in the young people with co-occurring intellectual disabilities and autism. This literature is difficult to summarise overall as, as well as having small sample sizes, the participants were not drawn from representative populations.

A further study had the advantage of being population based but was still small in size.[14] It compared the prevalence, and incidence, of mental ill-health in 77 adults with co-occurring intellectual disabilities and autism with 946 adults with intellectual disabilities without autism, and also with 154 individually age-matched, gender-matched, ability level matched and Down syndrome matched controls. The adults with autism had a higher point prevalence of problem behaviours than the 946 without autism, but compared with the 154 matched controls, there was no difference in prevalence or incidence of either problem behaviours or other mental ill-health.[14] Three large whole population studies using the Scotland's Census 2011 have reported that of people with intellectual disabilities, 21.7% reported mental health conditions,[3] and of people with autism, 33.0% of adults[6] and 7.6% of children[7] reported mental health conditions but did not report the rates for people with co-occurring intellectual disabilities and autism.

A recent study of Medicare claims data in Wisconsin, USA, by Bishop-Fitzpatrick and Rubenstein compared the physical and mental health conditions for adults aged 40–88 years old with a diagnosis of autism only (n=79) with those with a diagnosis of both autism and intellectual disabilities (n=64) between 2012 and 2015. The prevalence of chronic medical conditions was high among the entire sample, with elevated but not statistically significant prevalence rates for adults with both autism and intellectual disabilities on most conditions. However, ORs revealed a decreased likelihood of anxiety and depression for individuals with both autism and intellectual disabilities and a higher likelihood of epilepsy compared with those with autism only.[15] As the adults in this study were registered with Medicare, the sample may represent a

**Table 2** Prevalence of conditions in people with and without co-occurring intellectual disabilities and autism by age and sex

| Condition | Children/youth, 0–15 years N=2362 | | | Adults, 16–64 years N=3103 | | | Older people, 65+ years N=244 | | |
| --- | --- | --- | --- | --- | --- | --- | --- | --- | --- |
| | M N=1563 (100%) | F N=799 (100%) | Total N=2362 (100%) | M N=2073 (100%) | F N=1030 (100%) | Total N=3103 (100%) | M N=133 (100%) | F N=111 (100%) | Total N=244 (100%) |
| **People with co-occurring intellectual disabilities and autism** | | | | | | | | | |
| Mental health condition | 328 (21.0) | 152 (19.0) | 480 (20.3) | 768 (37.0) | 377 (36.6) | 1145 (36.9) | 80 (60.2) | 80 (72.1) | 160 (65.6) |
| Blindness/partial sight loss | 214 (13.7) | 177 (22.2) | 391 (16.6) | 355 (17.1) | 220 (21.4) | 575 (18.5) | 71 (53.3) | 80 (72.1) | 151 (61.9) |
| Deafness/partial hearing loss | 148 (9.5) | 95 (11.9) | 243 (10.3) | 301 (14.5) | 190 (18.4) | 491 (15.8) | 73 (54.9) | 81 (73.0) | 154 (63.1) |
| Physical disability | 618 (39.5) | 458 (57.3) | 1,076 (45.6) | 719 (34.7) | 508 (49.3) | 1 227 (39.5) | 86 (64.7) | 99 (89.2) | 185 (75.8) |

| Condition | Children/youth, 0–15 years N=913 969 | | | Adults, 16–64 years N=3 485 635 | | | Older people, 65+ years N=890 090 | | |
| --- | --- | --- | --- | --- | --- | --- | --- | --- | --- |
| | M N=467 543 (100%) | F N=446 426 (100%) | Total N=913 969 (100%) | M N=1 712 526 (100%) | F N=1 773 109 (100%) | Total N=3 485 635 (100%) | M N=383 606 (100%) | F N=506 484 (100%) | Total N=890 090 (100%) |
| **People without co-occurring intellectual disabilities and autism** | | | | | | | | | |
| Mental health condition | 1861 (0.4) | 980 (0.2) | 2841 (0.3) | 92 308 (5.4) | 95 108 (5.4) | 187 416 (5.4) | 14 760 (3.8) | 26 141 (5.2) | 40 901 (4.6) |
| Blindness/partial sight loss | 1793 (0.4) | 1439 (0.3) | 3232 (0.4) | 24 129 (1.4) | 16 954 (1.0) | 41 083 (1.2) | 30 389 (7.9) | 49 839 (9.8) | 80 228 (9.0) |
| Deafness/partial hearing loss | 2731 (0.6) | 2225 (0.5) | 4956 (0.5) | 70 543 (4.1) | 48 727 (2.7) | 119 270 (3.4) | 111 447 (29.1) | 114 393 (22.6) | 225 840 (25.4) |
| Physical disability | 3637 (0.8) | 2799 (0.6) | 6436 (0.7) | 81 655 (4.8) | 82 968 (4.7) | 164 623 (4.7) | 73 759 (19.2) | 109 103 (21.5) | 182 862 (20.5) |

F, female; M, male.

**Table 3** Independent predictors of mental health conditions in the whole population

| Characteristic | Regression 1 | | Regression 2 (including the interaction term: age × co-occurring intellectual disabilities and autism) | |
| --- | --- | --- | --- | --- |
| | OR | 95% CI | OR | 95% CI |
| Not present (reference) | – | – | – | – |
| Co-occurring intellectual disabilities and autism | 25.55 | 23.93 to 27.28 | 130.80 | 117.13 to 146.07 |
| Gender | | | | |
| Male (reference) | – | – | – | – |
| Female | 1.28 | 1.26 to 1.29 | 1.28 | 1.26 to 1.29 |
| Age | | | | |
| 0–15 (reference) | – | – | – | – |
| 16–24 | 9.45 | 9.04 to 9.88 | 11.32 | 10.78 to 11.89 |
| 25–34 | 21.17 | 20.29 to 22.09 | 25.31 | 24.13 to 26.53 |
| 35–44 | 29.73 | 28.50 to 31.00 | 35.49 | 33.87 to 37.20 |
| 45–54 | 29.19 | 27.98 to 30.44 | 34.84 | 33.24 to 36.51 |
| 55–64 | 22.63 | 21.63 to 23.62 | 27.00 | 25.75 to 28.30 |
| 65+ | 19.320 | 18.52 to 20.16 | 23.00 | 21.95 to 24.12 |
| Age × both intellectual disabilities and autism (years) | | | | |
| 0–15 (reference) | – | – | – | – |
| 16–24 | – | – | 0.15 | 0.13 to 0.18 |
| 25–34 | – | – | 0.10 | 0.08 to 0.12 |
| 35–44 | – | – | 0.08 | 0.07 to 0.10 |
| 45–54 | – | – | 0.09 | 0.07 to 0.11 |
| 55–64 | – | – | 0.12 | 0.09 to 0.15 |
| 65+ | – | – | 0.31 | 0.24 to 0.42 |
| Constant | | | | |
| – | 0.00 | – | 0.00 | – |

lower socioeconomic group than those with autism in the general population. Given the age of the sample, and the changes in diagnostic criteria for autism, this sample may not represent adults who are at the high functioning end of the spectrum.

With regards to sensory impairments, of the 36 matched youth with intellectual disabilities with and without autism, 38.9% with autism reported having visual problems compared with 50.0% without autism, and 13.9% with autism reported having hearing problems compared with 19.4% without autism.[12] An intellectual disabilities register study reported that 95 of the 368 (25.8%) adults with intellectual disabilities who had visual impairment also had markers for autism, compared with 422 of 2674 (16%) of those who had normal vision and that 46 of the 60 (76.7%) of the adults with intellectual disabilities and congenital blindness also had markers for autism compared with only 36 of the 67 (53.7%) with normal vision.[16]

We have not identified other papers on sensory impairments or any on physical disabilities in people with co-occurring intellectual disabilities and autism. However, previous large whole population studies that analysed data from the Scotland's Census 2011 have reported that of people with intellectual disabilities, 12.4% reported blindness/sight loss, 13.1% reported deafness/hearing loss and 32.6% reported physical disability.[3] Of people with autism, 12.1% of adults[6] and 3.5% of children reported blindness/sight loss,[7] 14.1% of adults[6] and 2.9% of children reported deafness/hearing loss[7] and 24.0% of adults[6] and 10.7% of children reported physical disability.[7] They did not, however, report the rates of these conditions for people with co-occurring intellectual disabilities and autism.

This high prevalence of health conditions among people with intellectual disabilities and people with autism spectrum disorders is partly attributable health inequalities[17 18] and to the fact that certain conditions, such as cerebral palsy, are associated with both intellectual and physical disabilities. Additionally, the socioeconomic status of individuals within these populations is typically lower than for members of the general population.[19 20] While prevalence rates of health conditions in a full country population for individuals with intellectual disabilities and individuals with autism have been compared with the general population in previous work

**Table 4** Independent predictors of blindness/partial sight loss in the whole population

| Characteristic | Regression 1 | | Regression 2 (including the interaction term: age × co-occurringintellectual disabilities and autism) | |
| --- | --- | --- | --- | --- |
| | OR | 95% CI | OR | 95% CI |
| Not present (reference) | – | – | – | – |
| Co-occurring intellectual disabilities and autism | 36.78 | 34.21 to 39.54 | 65.90 | 58.74 to 73.92 |
| Sex | | | | |
| Male (reference) | – | – | – | – |
| Female | 1.01 | 1.00 to 1.02 | 1.01 | 1.00 to 1.02 |
| Age (years) | | | | |
| 0–15 (reference) | – | – | – | – |
| 16–24 | 1.56 | 1.48 to 1.64 | 1.66 | 1.58 to 1.75 |
| 25–34 | 1.82 | 1.74 to 1.91 | 1.91 | 1.82 to 2.01 |
| 35–44 | 2.55 | 2.44 to 2.66 | 2.69 | 2.57 to 2.81 |
| 45–54 | 4.42 | 4.24 to 4.60 | 4.67 | 4.48 to 4.87 |
| 55–64 | 7.50 | 7.22 to 7.80 | 7.93 | 7.61 to 8.26 |
| 65+ | 31.06 | 29.95 to 32.22 | 32.75 | 31.51 to 34.04 |
| Age × both intellectual disabilities and autism (years) | | | | |
| 0–15 (reference) | – | – | – | – |
| 16–24 | – | – | 0.43 | 0.35 to 0.53 |
| 25–34 | – | – | 0.67 | 0.52 to 0.84 |
| 35–44 | – | – | 0.53 | 0.41 to 0.68 |
| 45–54 | – | – | 0.37 | 0.29 to 0.47 |
| 55–64 | – | – | 0.28 | 0.21 to 0.37 |
| 65+ | – | – | 0.25 | 0.19 to 0.33 |
| Constant | | | | |
| – | 0.00 | – | 0.00 | – |

using the Scotland's Census 2011,[3 4 7] no such study has been conducted on the prevalence of health conditions for those with both intellectual disabilities and autism.

The management of care and treatment plans for individuals with multiple health conditions or disabilities presents significant challenges for healthcare practitioners. A review of 123 studies on care management for individuals with multiple chronic conditions in the USA reported that these patients access services more frequently and use a larger range of services than other patients, making the coordination of their care more difficult and often leading to suboptimal care.[21] Existing evidence also suggests that when individuals with intellectual disabilities have additional long-term conditions, these conditions are more poorly managed than for members of the general population with the same conditions.[22]

Given the frequent overlap of intellectual disabilities and autism, information on the associated coexisting conditions is important to assist policy makers, planners and practitioners to best adapt services for individuals with co-occurring intellectual disabilities and autism. This paper aims to investigate the prevalence of mental health conditions, sensory impairments and physical disability in children, adults and older adults with co-occurring intellectual disabilities and autism, compared with other people.

## METHODS
### Data source
Scotland's Census provides information on Scotland's population every 10 years, with the most recent Census on 27 March 2011.[23] The Census provides information on the number and characteristics of Scotland's population and households on the Census date.

It is a legal requirement to complete the census form, and households were informed that failure to make a Census return or supplying false information could result in a £1000 fine. A very high response rate was achieved, with an estimated 94% of all of Scotland's population completing the census. The Census team used a Census Coverage Survey with about 40 000 households to estimate numbers and characteristics of the missing 6%.[24] The Coverage Survey and Census records were deterministically matched to check for duplicates. Individuals estimated to have been missed were then imputed

using a subset of characteristics from real individuals. The edit and imputation methodology was adapted from the Office for National Statistics' rigorous and systematic guidelines.[24]

The Census was completed by the head of each household on behalf of all occupants of the household. We consider it unlikely that people with co-occurring intellectual disabilities and autism completed the form, given the reading age required to do so. Rather, we expect that the people who completed the form on their behalf were parent-carers in family households, support workers for people living in supported accommodation and the managers/key workers at communal establishments.

## Variables

The census included a question to identify people with intellectual disabilities and autism, as well as mental health conditions, sensory impairments and physical disabilities: 'Do you have any of the following conditions which have lasted, or are expected to last, at least 12 months? Tick all that apply'. There was a choice of 10 possible responses to this question: deafness or partial hearing loss; blindness or partial sight loss; learning disability (eg, Down's syndrome); learning difficulty (eg, dyslexia); developmental disorder (eg, autistic spectrum disorder or Asperger's syndrome); physical disability; mental health condition; long-term illness, disease or condition; and other conditions. For 'other conditions', the option of providing more detail in an open-text response was provided.

In Scotland, the term 'learning disability' is used synonymously with that of 'intellectual disabilities' used internationally. Importantly, the Census differentiated between intellectual disabilities and specific learning disabilities, and between intellectual disabilities and autism.

During the methodology development for Scotland's Census 2011, cognitive question testing was undertaken on the questions on long-term health conditions and disabilities. This was to determine whether the questions were answered accurately and to identify any changes needed to improve data quality and/or the acceptability of the way questions were phrased. Cognitive interviewing is a widely used approach to critically evaluate and improve survey questionnaires.[25] This approach enables researchers to modify survey material to enhance clarity. Retrospective probing was conducted with 102 participants with a variety of sex, age and health conditions and disabilities (including people with more than one of the conditions). They included people with autism, intellectual disabilities, dyslexia, dyspraxia, speech impairment, mental health conditions (both milder and more serious), and other long-term conditions.[23] Using the cognitive interviewing results, the question aimed to detect autism was improved and rephrased to better capture this information. The questions on intellectual disability, mental health condition, visual impairment, hearing impairment and physical disability did not require any modifications.[26]

The Census team imputed answers for the 14.7% who did not tick any of the boxes in question on long-term conditions, based on their free-text answers for this question, and answers to other health questions in the Census, which increased the completion rate to 97.4%. For the remaining 2.6%, the Census team assumed the most plausible explanation was that the person had no long-term condition but did not see the 'No condition' check box at the end of the question, and hence recorded them as having no conditions.

## Data analysis

We calculated the number and rate per 1000 population of children and adults with co-occurring intellectual disabilities and autism. We then calculated the number and percentage of people with mental health conditions, deafness or partial hearing loss, blindness or partial sight loss and physical disability, for those with co-occurring intellectual disabilities and autism, compared with individuals who do not have co-occurring intellectual disabilities and autism using $\chi^2$ tests. For the whole population, we then used logistic regression to calculate the ORs (OR: 95% CI) of co-occurring intellectual disabilities and autism statistically predicting the binary response of having each of the four specific types of long-term health conditions, adjusted for age and sex. Sex was binary, with males being the reference group. Age was categorised into groups: 0–15, 16–24, 25–34, 35–44, 45–54, 55–64, 65–74 and 75+, with 0–15 years as the reference group. We repeated the regressions, including the interaction term of age x co-occurring intellectual disabilities and autism, as people with the most severe disabilities die earlier, which may affect the profile of additional health problems differently to that seen in the general population. The same reference groups were used. All analysis was conducted using SPSS software V.22.

## Patient and public involvement

The Scottish Learning Disabilities Observatory, where this research was undertaken, has a specific remit for people with intellectual disabilities and people with autism. Its steering group includes partners from third sector organisations and experts by experience who approved the workplan for this project prior to it commencing. Results from this study will be disseminated for people with intellectual disabilities and autism in an easy-read version via the Scottish Learning Disabilities Observatory website, newsletters and conference.

## RESULTS
### Characteristics of the sample

Scotland's Census 2011 includes records on 5 295 403 people aged 0–75+ years. A total of 5709 of 5 295 403 (1.08/1000) people had co-occurring intellectual disabilities and autism,

of whom 3769 (66.0%) were male and 1940 (44.0%) were female. Overall, 2362/916 331 (2.58/1000) of the total population of children (0–15 years), and 3347/4 379 072 (0.76/1000) adults (16–75+ years) had co-occurring intellectual disabilities and autism.

Compared with the population who did not have co-occurring intellectual disabilities and autism, the population with co-occurring intellectual disabilities and autism had more males (66.0% vs 48.5%; $\chi^2$=703.5; df=1; p<0.001); were younger ($\chi^2$=3894.7; df=7; p<0.001); were more likely to have been born in the UK rather than elsewhere ($\chi^2$=101.9; df=1; p<0.001), revealing lesser geographic mobility; and were no different with regards to Caucasian versus non-Caucasian ethnicity ($\chi^2$=1.1; df=1; p=0.3) (table 1).

### Long-term health conditions

Table 2 shows the proportion of people with co-occurring intellectual disabilities and autism, who had each of the four additional long-term health conditions, compared with people who did not have co-occurring intellectual disabilities and autism. Some people in the sample had more than one long-term health condition.

### Mental health condition

Adjusting for age and sex, given the different distributions compared with the general population, having co-occurring intellectual disabilities and autism had an OR=25.55 (95% CI 23.93 to 27.28) in predicting mental health conditions (table 3). When the interaction term was added (age × co-occurring intellectual disabilities and autism), co-occurring intellectual disabilities and autism had an OR=130.80 (95% CI 117.13 to 146.07) in predicting a mental health condition (table 3).

### Blindness or partial sight loss

Adjusting for age and sex, having co-occurring intellectual disabilities and autism had an OR=36.78 (95% CI 34.21 to 39.54) in predicting blindness or partial sight loss (table 4). When the interaction term was added (age × co-occurring intellectual disabilities and autism), co-occurring intellectual disabilities and autism had an OR=65.90 (95% CI 58.74 to 73.92) in predicting blindness or partial sight loss (table 4).

### Deafness or partial hearing loss

Adjusting for age and sex, having co-occurring intellectual disabilities and autism had an OR=11.33 (95% CI 10.43 to 12.31) in predicting deafness or partial hearing loss (table 5). When the interaction term was added (age × co-occurring intellectual disabilities and autism), co-occurring intellectual disabilities and autism had an OR=22.00 (95% CI 19.20 to 25.21) in predicting deafness or partial hearing loss (table 5).

### Physical disability

Adjusting for age and sex, having co-occurring intellectual disabilities and autism had an OR=61.16 (95% CI 57.60 to 64.94) in predicting physical disability (table 6). When the interaction term was added (age × co-occurring

intellectual disabilities and autism), co-occurring intellectual disabilities and autism had an OR=157.54 (95% CI 144.58 to 171.66) in predicting physical disability (table 6).

## DISCUSSION
### Principle findings

Mental health conditions, blindness or partial sight loss, deafness or partial hearing loss and physical disability were all significantly more common in people with co-occurring intellectual disabilities and autism than people without these co-occurring conditions. The ORs after adjusting for age and sex and the interaction term were substantial, being 131, 66, 22 and 158, respectively. This is important as each of these conditions are disabling and can significantly impact an individual's quality of life. They contribute to high rates of multimorbidity which, on top of communication and cognitive problems due to autism and intellectual disabilities, renders assessments, diagnosis, and treatment of additional health problems more complex than for other people.

Across all age groups, blindness, deafness and physical disability were more common in women than men with co-occurring intellectual disabilities and autism, unlike the gender ratios in people without co-occurring intellectual disabilities and autism. Mental health conditions were more common in men than women with co-occurring intellectual disabilities and autism, except for the 65+ year group, contrary to the gender ratios in other people. All conditions were more prevalent with increasing age in the people with co-occurring intellectual disabilities and autism, except for physical disability, which was more common in the children/youth and older people than in the adults.

### Comparison with existing literature

The prevalence of these additional long-term health conditions has seldom been investigated in people with co-occurring intellectual disabilities and autism, particularly in comparison with other people, and never, to our knowledge, as a total population study. All of the long-term health conditions were more common than in those without co-occurring intellectual disabilities and autism.

Smaller, less representative studies have reported a higher rate of mental health conditions in adults and youth with co-occurring intellectual disabilities and autism compared with those with intellectual disabilities and without autism,[9–13] but not all.[14] People with autism have been reported to have more mental health conditions than other people (OR=9 in adults and OR=16 in children),[6 7] as have people with intellectual disabilities compared with other people (OR=7),[3] using the same Scotland's Census 2011 data as in this current paper, whereas the comparable ratio we now report for people with co-occurring intellectual disabilities and autism for mental health conditions was OR=26. Having the co-occurring conditions therefore presents a much higher

**Table 5** Independent predictors of deafness/partial hearing loss in the whole population

| Characteristic | Regression 1 | | Regression 2 (including the interaction term: age × co-occurring intellectual disabilities and autism) | |
|---|---|---|---|---|
| | OR | 95% CI | OR | 95% CI |
| Not present (reference) | – | – | – | – |
| Co-occurring intellectual disabilities and autism | 11.33 | 10.43 to 12.31 | 22.00 | 19.20 to 25.21 |
| Gender | | | | |
| Male (reference) | – | – | – | – |
| Female | 0.69 | 0.68 to 0.69 | 0.69 | 0.68 to 0.69 |
| Age | | | | |
| 0–15 (reference) | – | – | – | – |
| 16–24 | 1.56 | 1.50 to 1.62 | 1.59 | 1.52 to 1.65 |
| 25–34 | 2.36 | 2.27 to 2.45 | 2.41 | 2.32 to 2.50 |
| 35–44 | 4.24 | 4.10 to 4.38 | 4.35 | 4.20 to 4.50 |
| 45–54 | 8.55 | 8.29 to 8.82 | 8.77 | 8.50 to 9.05 |
| 55–64 | 18.76 | 18.20 to 19.34 | 19.24 | 18.66 to 19.85 |
| 65+ | 69.65 | 67.63 to 71.72 | 71.38 | 69.27 to 73.55 |
| Age × both intellectual disabilities and autism (years) | | | | |
| 0–15 (reference) | – | – | – | – |
| 16–24 | – | – | 0.63 | 0.51 to 0.79 |
| 25–34 | – | – | 0.74 | 0.57 to 0.95 |
| 35–44 | – | – | 0.39 | 0.30 to 0.52 |
| 45–54 | – | – | 0.29 | 0.22 to 0.38 |
| 55–64 | – | – | 0.22 | 0.16 to 0.29 |
| 65+ | – | – | 0.22 | 0.17 to 0.30 |
| Constant | | | | |
| – | 0.01 | – | 0.01 | – |

risk of mental health conditions than either intellectual disabilities or autism on their own.

The previous small study of youth reported lower rates of visual and hearing impairments in those with co-occurring intellectual disabilities and autism (38.9% and 13.9%) compared with those with intellectual disabilities but without autism (50% and 19.4%).[12] This was in contrast with the larger study reporting more autistic symptoms in adults with intellectual disabilities and visual impairments than in adults with intellectual disabilities but without visual impairments.[16] Adults with autism have been reported to have more blindness or partial sight loss, and deafness or partial hearing loss than other people (12.1% and 17.5%),[6] as have children with autism (3.5% and 2.9%),[7] and people (children and adults combined) with intellectual disabilities compared with other people (13.1% and 12.4%),[3] using the same data from Scotland's Census 2011 as in this current paper. This current study found the comparable rates for people with co-occurring intellectual disabilities and autism for blindness or partial sight loss, and deafness or partial hearing loss was 21.7% and 19.3% for adults, and 16.6% and 10.3% for children. Having the co-occurring conditions of intellectual disabilities and autism therefore presents a much higher risk of sensory impairments than for children and adults with autism and for people with intellectual disabilities (although children were not separately studied in the previous report).

Regarding physical disability, 32.6% of people with intellectual disabilities were previously reported to have physical disability using the same dataset as the current study.[3] Of people with autism, 24.0% of adults and 10.7% of children reported physical disability in this dataset.[6 7] These rates are lower than those we report in this current study of people with co-occurring intellectual disabilities and autism—45.6% of children and 42.2% of adults.

### Strengths and limitations
Strengths of the study include its large scale and general population comparison group, whole population coverage and very high response rate so the results are representative of the whole population. Intellectual disabilities, autism and the long-term health conditions were enquired about systematically for everyone in the population. We believe the results are therefore generalisable to other high-income countries. The cognitive question testing

**Table 6** Independent predictors of physical disability in the whole population

| | Regression 1 | | Regression 2 (including the interaction term: age × co-occurringintellectual disabilities and autism) | |
| --- | --- | --- | --- | --- |
| Characteristic | OR | 95% CI | OR | 95% CI |
| Not present (reference) | – | – | – | – |
| Co-occurring intellectual disabilities and autism | 61.16 | 57.60 to 64.94 | 157.54 | 144.58 to 171.66 |
| Gender | | | | |
| Male (reference) | – | – | – | – |
| Female | 1.06 | 1.06 to 1.07 | 1.06 | 1.06 to 1.07 |
| Age | | | | |
| 0–15 (reference) | – | – | – | – |
| 16–24 | 1.44 | 1.39 to 1.50 | 1.57 | 1.51 to 1.64 |
| 25–34 | 2.60 | 2.52 to 2.69 | 2.86 | 2.76 to 2.96 |
| 35–44 | 5.87 | 5.70 to 6.04 | 6.47 | 6.27 to 6.67 |
| 45–54 | 10.61 | 10.31 to 10.91 | 11.66 | 11.32 to 12.02 |
| 55–64 | 20.73 | 20.17 to 21.31 | 22.76 | 22.10 to 23.44 |
| 65+ | 43.68 | 42.52 to 44.88 | 47.89 | 46.53 to 49.30 |
| Age × both intellectual disabilities and autism (years) | | | | |
| 0–15 (reference) | – | – | – | – |
| 16–24 | – | – | 0.41 | 0.36 to 0.47 |
| 25–34 | – | – | 0.34 | 0.29 to 0.41 |
| 35–44 | – | – | 0.12 | 0.10 to 0.15 |
| 45–54 | – | – | 0.08 | 0.06 to 0.09 |
| 55–64 | – | – | 0.04 | 0.03 to 0.06 |
| 65+ | – | – | 0.05 | 0.040 to 0.07 |
| Constant | | | | |
| – | 0.01 | – | 0.01 | – |

during the design of the Census is a further strength. The Census had clear categories to distinguish between intellectual disabilities, specific learning disability (like dyslexia) and autism.

Limitations include the proxy reporting, which may or may not reflect self-reports. However, without proxy reports, we would have no information on people unable to self-report due to their disabilities, and a previous review on the topic concluded that overall, proxy reports are a useful addition to determine aspects of well-being in people with intellectual disabilities.[27] Additionally, people were reported who were known to have autism/Asperger's syndrome, intellectual disabilities and the additional long-term health conditions, rather than detailed individual research assessments being undertaken, which are clearly not possible in such large population studies, and may therefore be subject to a degree of error, which we were not able to check. Individuals who were known to have intellectual disabilities and autism are higher healthcare users and so are more likely to receive a diagnosis for other healthcare conditions than members of the general population who do not access healthcare services as frequently. It is also important to note that the Scotland's

Census 2011 was administered 8 years ago, and so any potential changes in prevalence rates of the conditions investigated in this paper are not captured by this analysis.

## Implications

There is a greater than double disadvantage for people with co-occurring intellectual disabilities and autism, in terms of additional long-term health conditions. We found that, and quantified, the extent to which mental health conditions, sensory impairments and physical disabilities are more common than in people without co-occurring intellectual disabilities and autism and in people with just intellectual disabilities or just autism. This may well impact on quality of life. It raises challenges for staff working with people with these co-occurring conditions in view of the additional complexity in assessments, diagnoses and interventions, as sensory impairments and mental health conditions in particular interact with the person's pre-existing communication and cognitive problems in this context. Therefore, it is important that these co-occurring conditions are planned for with staff being trained, equipped, resourced and prepared to address the challenge.

**Acknowledgements** We would like to thank the National Records of Scotland Census team and the Scottish population who completed the Census.

**Contributors** KD analysed and interpreted the data and wrote the first draft of the manuscript. ER contributed to data access, data interpretation and drafting the manuscript. MF provided advice on response to reviewer comments regarding choice of statistical methods and contributed to the writing of the manuscript. S-AC conceived and managed the project, interpreted data and contributed to writing the manuscript. All approved the final version of the manuscript. S-AC is the study guarantor.

**Funding** This work was supported by the UK Medical Research Council, grant number: MC_PC_17217) and the Scottish Government via the Scottish Learning Disabilities Observatory.

**Disclaimer** The study sponsor and funders had no role in the study design; in the collection, analysis and interpretation of data; in the writing of the report; and in the decision to submit the article for publication. The researchers are independent from the funders.

**Competing interests** None declared.

**Patient and public involvement** Patients and/or the public were involved in the design, or conduct, or reporting, or dissemination plans of this research. Refer to the Methods section for further details.

**Patient consent for publication** Not required.

**Provenance and peer review** Not commissioned; externally peer reviewed.

**Data availability statement** Data are available in a public, open access repository. Data are available via National Records of Scotland, following project approval. Data are available at the following link https://www.scotlandscensus.gov.uk/ods-web/data-warehouse.html#additionaltab.

**ORCID iDs**
Kirsty Dunn http://orcid.org/0000-0003-1701-2190
Sally-Ann Cooper http://orcid.org/0000-0001-6054-7700

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
