## [Reviewer comments · BMJ Open]

ARTICLE DETAILS

TITLE (PROVISIONAL)	The prevalence of mental health conditions, sensory impairments, and physical disability in people with co-occurring intellectual disabilities and autism compared with other people – a cross-sectional total population study in Scotland
AUTHORS	Dunn, Kirsty; Rydzewska, Ewelina; Fleming, Michael; Cooper, Sally-Ann

VERSION 1 – REVIEW

REVIEWER	Eric Rubenstein UW Madison, USA
REVIEW RETURNED	02-Nov-2019

GENERAL COMMENTS	The authors have a large and unique data set, a full Scottish census, and assess intellectual disability and autism. The authors have a long track record of publishing these data, which makes sense given the 94% response rate. I struggle with the analytic design of this paper, both in concept and methodology. 1. Why is the comparison group the typically developing population? We know through all of our studies that people with ID and ASD have more health conditions than the typical developing population. Further, a lot of that is at least partially attributable to disparity and socioeconomic status. By comparing ID/ASD to TD the results are still greatly confounded by SES. Further, some of the conditions you look at are intrinsic to ID. For example, ID is associated with cerebral palsy which is a cause of physical disability- that really isn't a 'co-occurring condition', that is the condition. I understand that the data did not have the detail to look deeper than these broader categories, but all categories could include the phenotype of the ID/ASD. 2. This is a cross sectional study and should be emphasized more throughout. Further, since you have all data I highly recommend using prevalence difference instead of odds ratios as your outcome measure. The odds ratios are wildly high because the prevalence in the typically developing population is so low. Prevalence difference allows you to better display the true difference, not a boosted ratio because of low prevalence. You could use linear risk regression. Rates are not appropriate since the data are cross-sectional. Additionally, your interpretation of your regression models lead into the Table 2 fallacy (https://www.ncbi.nlm.nih.gov/pubmed/23371353). Those regression coefficients are confounded since you did not adjust for the confounders of the age-health condition or sex-health condition association. With your cross-sectional data, it is inaccurate to use the word 'predicting'
--

	Minor points The title is very wordy and too detailed The literature review focuses on smaller studies, see Bishop and Rubenstein 2019 for a population based comparison between adults with ASD and adults with ASD+ID. The sample is now 8 years ago, you should address potential changes over time. I think you should also consider that people with ID/ASD are higher health care users and more likely to get diagnoses for things
--	--

REVIEWER	Radosław Wolniak Silesian University of Technology
REVIEW RETURNED	05-Nov-2019

GENERAL COMMENTS	The paper is good but Authors should increase the amount of literature position used in the background section. This part is rather the weakest part of the paper. Authors should add more in-deep literature review and then in discussion describe what are the links between the literature and the conducted research.
--

REVIEWER	Manjula Marella The University of Melbourne
REVIEW RETURNED	29-Nov-2019

GENERAL COMMENTS	This was an interesting paper to read about a specific group of people with co-occurring intellectual disabilities and autism. The paper provides strong evidence that this group is at a high risk of having other impairments compared to others. Although self-reported responses to Census questions were used for this sample, the questions have been cognitively tested. The analysis of the data and presentation of findings is well done. The authors have compared their findings with the existing literature. I would have liked to read some more literature around implications for services and interventions for people with multiple disabilities as part of the justification for this research targeting this specific group. Similarly, discussion section could elaborate how findings from this paper can be linked with the existing policies and practices in Scotland. The title is too wordy and I suggest revising it to something short and catchy.
--

VERSION 1 – AUTHOR RESPONSE

Reviewer: 1

Reviewer Name: Eric Rubenstein

Institution and Country: UW Madison, USA

Please state any competing interests or state 'None declared': none declared

Please leave your comments for the authors below

The authors have a large and unique data set, a full Scottish census, and assess intellectual disability and autism. The authors have a long track record of publishing these data, which makes sense given

the 94% response rate. I struggle with the analytic design of this paper, both in concept and methodology.

1. Why is the comparison group the typically developing population? We know through all of our studies that people with ID and ASD have more health conditions than the typical developing population. Further, a lot of that is at least partially attributable to disparity and socioeconomic status. By comparing ID/ASD to TD the results are still greatly confounded by SES. Further, some of the conditions you look at are intrinsic to ID. For example, ID is associated with cerebral palsy which is a cause of physical disability- that really isn't a 'co-occurring condition', that is the condition. I understand that the data did not have the detail to look deeper than these broader categories, but all categories could include the phenotype of the ID/ASD.

Response: We have added text in the discussion on these points: "While prevalence rates of mental health conditions and impairments in a full country population for individuals with intellectual disabilities, and separately for individuals with autism have been compared to the general population, no such study has been conducted on the prevalence of mental health conditions and impairments for those with co-occurring intellectual disabilities and autism." (page 16)

"Cerebral palsy is associated with intellectual disabilities, but the extent of this association does not appear to account for the differences found in physical disability: cerebral palsy has been reported to occur in 13% of children and young people with intellectual disabilities,²¹ and in 3.2% of general population children and young people in the USA.²²" (page 17)

In the general population, a gradient across socioeconomic status is typically associated with a gradient across most measures of health. In autistic people who do not have intellectual disabilities, socioeconomic status appears to be associated with health. Scotland's Census, 2011 did not enquire about income, hence neighbourhood deprivation (based on post-code) would typically be taken as a marker for socioeconomic status. However, in our study, all of the exposed group have intellectual disabilities (as well as autism), and studies of adults with intellectual disabilities repeatedly do not find associations of a gradient in extent of neighbourhood deprivation with measures of health, for several complex reasons. For examples, adults with intellectual disabilities living with their parents may have skilled professional parents and live in affluent areas, whilst other adults with intellectual disabilities may live in congruent care setting with paid carers shared by many residents but with the location also being in an affluent area as that is where large-sized housing often is located. In contrast, an adult may rent a flat and have a 24 hour supported living package shared with just one other person (ie individualised care) and have frequent visits from and stays with skilled professional parents, but with their flat being in one of the most deprived areas as that is where housing association accommodation tends to be located. In our study, all of our exposed group had intellectual disabilities, so a neighbourhood deprivation effect on health is not expected. Hence we have added: "There are likely to be many biological, social, and environmental reasons accounting for these results." (page 15). If advised by the editor we will expand upon and reference this more fully, but have not done so in this revision, to not detract from the main thrust of the paper.

Unfortunately, we do not have information on the underlying causes of intellectual disabilities and autism in the dataset, so cannot add this information.

2. This is a cross sectional study and should be emphasized more throughout. Further, since you have all data I highly recommend using prevalence difference instead of odds ratios as your outcome measure. The odds ratios are wildly high because the prevalence in the typically developing population is so low. Prevalence difference allows you to better display the true difference, not a boosted ratio because of low prevalence. You could use linear risk regression. Rates are not appropriate since the data are cross-sectional.

Response: We agree, and have now more prominently emphasised that this is a cross-sectional study in the title, abstract, strengths and limitations bullet points, methods, results, and limitations in the discussion.

A statistician in biomedical sciences is now a co-author of the paper, and we have revised the presentation of the statistics. We have added examples of prevalence differences (page 14):

"Prevalence differences are apparent from table 2. For children and young people aged 0-15 years,

the difference in prevalence between the population with co-occurring intellectual disabilities and autism, and the rest of the general population, was: 20.0/100 for mental health conditions, 16.2/100 for blindness/partial sight loss, 9.8/100 for deafness/partial hearing loss, and 44.9/100 for physical disability. For adults aged 16-64 years, the difference in prevalence between the population with co-occurring intellectual disabilities and autism, and the rest of the general population, was: 31.5/100 for mental health conditions, 17.3/100 for blindness/partial sight loss, 12.4/100 for deafness/partial hearing loss, and 34.8/100 for physical disability. For older adults aged 65+ years, the difference in prevalence between the population with co-occurring intellectual disabilities and autism, and the rest of the general population, was: 61.0/100 for mental health conditions, 52.9/100 for blindness/partial sight loss, 37.7/100 for deafness/partial hearing loss, and 55.3/100 for physical disability.” We have also added an acknowledgement in the limitations of the discussion that odds ratios may overestimate the strength of association (page 19): “We acknowledge that the use of odds ratios may overestimate the strength of associations in cross-sectional studies where the prevalence in the general population is very low; calculating odds ratios has enabled us to draw comparisons with previously published results on health and disabilities of people with intellectual disabilities and on people with autism. We also present example prevalence differences.” We have not undertaken linear regression, as the prevalences are very small hence the RRs and ORs would be reasonably similar.

Additionally, your interpretation of your regression models lead into the Table 2 fallacy (<https://www.ncbi.nlm.nih.gov/pubmed/23371353>). Those regression coefficients are confounded since you did not adjust for the confounders of the age-health condition or sex-health condition association. With your cross-sectional data, it is inaccurate to use the word 'predicting'

Response: We agree, and have made amendments accordingly. In the methods and results (pages 10 and 12-13) we have removed the term “predicting”. We have renamed tables 3-6 to e.g. “Effect of co-occurring intellectual disabilities and autism on mental health conditions in the whole population, adjusted for sex and age”. We have added into the discussion (page 19) “It is also important to note that whilst our regressions adjusted for sex and age, the effect sizes for sex and age shown in tables 3-6 might not be the total effect of sex and age on the four outcomes: they show the proportion of the sex, age effect on the odds ratio for the four outcomes that are not mediated through any sex, age effect on co-occurring intellectual disabilities and autism.²³” We have also added the reference the reviewer provided as it is a helpful explanation of this point.

Minor points

The title is very wordy and too detailed

Response: The first part of the title has been shortened (and location and study type added, in keeping with the editor’s instructions).

The literature review focuses on smaller studies, see Bishop and Rubenstein 2019 for a population based comparison between adults with ASD and adults with ASD+ID.

Response: We have added Bishop and Rubenstein, 2019 to the literature review (page 5).

The sample is now 8 years ago, you should address potential changes over time.

Response: We agree and have added to the limitations (page 19): “Scotland’s Census, 2011 was administered 8 years ago, and so any potential changes in prevalence of conditions since then are not captured by our analyses.”

I think you should also consider that people with ID/ASD are higher health care users and more likely to get diagnoses for things

Response: We agree this this should be considered, and have added the following (page 19): “People with intellectual disabilities and autism are higher health care users than other people and so may receive more diagnoses, but they are also subject to “diagnostic overshadowing”. We do not know the extent to which these factors may impact on reporting of mental health conditions, sensory

impairments, and physical disability at Scotland's Census, 2011. However, given the long-term nature of these conditions, any impact is likely to be less than it would be for acute conditions."

Reviewer: 2

Reviewer Name: Radosław Wolniak

Institution and Country: Silesian University of Technology

Please state any competing interests or state 'None declared': None declared

Please leave your comments for the authors below

The paper is good but Authors should increase the amount of literature position used in the background section. This part is rather the weakest part of the paper. Authors should add more in-deep literature review and then in discussion describe what are the links between the literature and the conducted research.

Response: The quantity and depth of the literature review in the introduction reflects the shortage of studies which have previously investigated mental health conditions, physical disability, and/or sensory impairments among the population with co-occurring intellectual disabilities and autism. We agree it is limited, perhaps highlighting the need for our study. In the discussion we have drawn comparison with the existing literature as far as we believe that to be possible. Reviewer 1 signposted us to an additional recent relevant paper (Bishop & Rubenstein, 2019) which we have added (page 5).

Reviewer: 3

Reviewer Name: Manjula Marella

Institution and Country: The University of Melbourne

Please state any competing interests or state 'None declared': None declared

Please leave your comments for the authors below

This was an interesting paper to read about a specific group of people with co-occurring intellectual disabilities and autism. The paper provides strong evidence that this group is at a high risk of having other impairments compared to others. Although self-reported responses to Census questions were used for this sample, the questions have been cognitively tested. The analysis of the data and presentation of findings is well done. The authors have compared their findings with the existing literature. I would have liked to read some more literature around implications for services and interventions for people with multiple disabilities as part of the justification for this research targeting this specific group. Similarly, discussion section could elaborate how findings from this paper can be linked with the existing policies and practices in Scotland.

Response: Existing literature on implications for services have been added to the clinical implications section of the discussion (page 19), as have the links and relevance to Scottish policy and strategy (page 20).

The title is too wordy and I suggest revising it to something short and catchy.

Response: The title has been amended.

VERSION 2 – REVIEW

REVIEWER	Eric Rubenstein University of Wisconsin Madison, USA
REVIEW RETURNED	12-Feb-2020

GENERAL COMMENTS	Thank you for addressing my concerns. Minor points -Can you make it clear that the comparison to the autistic-only and ID-only were from the same data source so can be compared? That is not clear in the text. -Too many significant figures are presented for odds ratio and CIs. -Avoid using the word 'comorbid' as some may take offense to implying id or asd is a morbidity
---

REVIEWER	Radosław Wolniak Silesian University of Technology Organization and Management Department
REVIEW RETURNED	24-Feb-2020

GENERAL COMMENTS	There is a small amount of position in theoretical background of the paper. The authors should use more relevant papers from good peer reviewed Journals. On this basis, they should prepare a more in-depth theoretical background of the paper.
---

REVIEWER	Manjula Marella The University of Melbourne, Australia
REVIEW RETURNED	28-Feb-2020

GENERAL COMMENTS	The authors have addressed all comments from the reviewers in this version.
---

VERSION 2 – AUTHOR RESPONSE

Reviewer: 1

Reviewer Name: Eric Rubenstein

Institution and Country: University of Wisconsin Madison, USA Please state any competing interests or state 'None declared': none

Please leave your comments for the authors below Thank you for addressing my concerns.

Minor points

-Can you make it clear that the comparison to the autistic-only and ID-only were from the same data source so can be compared? That is not clear in the text.

Response: The following text has been included to clarify this:

'A whole population study using the Scotland Census 2011 reported that 21.7% of people with intellectual disabilities also had autism, and 18.0% of people with autism also had intellectual disabilities. (page 4)

'Three large whole population studies using the Scotland Census 2011 have reported that of people with intellectual disabilities, 21.7% reported mental health conditions; and of people with autism, 33.0% of adults,⁶ and 7.6% of children⁷ reported mental health conditions' (page 5)

'However, previous large whole population studies which analysed data from the Scotland Census 2011 have reported that of people with intellectual disabilities...' (page 7)

'While prevalence rates of health conditions in a full country population for individuals with intellectual disabilities and individuals with autism have been compared to the general population in previous work using the Scotland Census 2011...' (page 7)

'People with autism have been reported to have more mental health conditions than other people (OR=9 in adults and OR=16 in children), as have people with intellectual disabilities compared with other people (OR=7), using the same Scotland's Census, 2011 data as in this current paper,...' (page 16).

'Adults with autism have been reported to have more blindness or partial sight loss, and deafness or partial hearing loss than other people (12.1% and 17.5%),⁶ as have children with autism (3.5% and 2.9%),⁷ and people (children and adults combined) with intellectual disabilities compared with other people (13.1% and 12.4%),³ using the same data from Scotland's Census 2011 as in this current paper.' (page 17)

'Regarding physical disability, 32.6% of people with intellectual disabilities were previously reported to have physical disability using the same dataset as the current study. Of people with autism, 24.0% of adults and 10.7% of children reported physical disability in this dataset.' (page 17)

-Too many significant figures are presented for odds ratio and CIs.

Response: We have adjusted the tables and figures in the text to display only two significant figures for odds ratios and CIs (pages 13-15, 28-30)

-Avoid using the word 'comorbid' as some may take offense to implying id or asd is a morbidity

Response: We have replaced the word comorbid with 'co-existing' throughout the manuscript (pages 4, 5, 8).

Reviewer: 2

Reviewer Name: Radosław Wolniak

Institution and Country:

Silesian University of Technology

Organization and Management Department

Please state any competing interests or state 'None declared': None declared

Please leave your comments for the authors below

There is a small amount of position in theoretical background of the paper. The authors should use more relevant papers from good peer reviewed Journals. On this basis, they should prepare a more in-depth theoretical background of the paper.

Response: The authors have been unable to identify any further relevant background literature on this topic from good peer reviewed journals. The lack of previous literature on this topic points to the importance of our paper.

Reviewer: 3

Reviewer Name: Manjula Marella

Institution and Country: The University of Melbourne, Australia Please state any competing interests

or state 'None declared': None declared

Please leave your comments for the authors below

The authors have addressed all comments from the reviewers in this version.